# The relationship between family conflict resolution methods and depressive symptoms in patients with chronic diseases

**Min Jeong Joo[1,2], Jisu Ko[1,2], Jae Hyeok Lim[1,2], Dan Bi Kim[1,2], Eun-Cheol Park**◉[2,3]*

**1** Department of Public Health, Graduate School, Yonsei University, Seoul, Republic of Korea, **2** Institute of Health Services Research, Yonsei University, Seoul, Republic of Korea, **3** Department of Preventive Medicine, Yonsei University College of Medicine, Seoul, Republic of Korea

* ECPARK@yuhs.ac

## Abstract

### Background

Individuals with chronic diseases are more sensitive to depressive symptoms and stress compared to the general population. The complexity and unpredictability of these diseases necessitate family involvement in their management. However, long-term disease can exhaust both patients and their families, leading to conflicts and increased stress, thus exacerbating depressive symptoms. This longitudinal study investigated the impact of family conflict resolution methods on depressive symptoms among chronic disease patients in Korea.

### Methods

We used data from the Korean Welfare Panel Study, collected from 2012 to 2022, analyzing 10,969 chronically ill cohabiting or married individuals. Chi-square tests were used to compare group characteristics, and generalized estimating equation models were used for regression analysis, focusing on Center for Epidemiologic Studies Depression Scale-11 scores, family conflict resolution changes, and covariates.

### Results

Participant groups that changed from positive to negative conflict resolution methods were more likely to have depressive symptoms than the group that did not change from positive methods (positive → negative odds ratios (OR) = 1.34, confidence intervals (CI) = 1.24–1.44). In addition, participants who did not change from negative methods were significantly more depressed than those who maintained positive methods over time (negative → negative OR = 1.48, 95% CI = 1.37–1.59). Uncollaborative discussions and domestic violence resolution methods were related to depressive symptoms in family conflict resolution methods.

### Conclusion

Negative family conflict resolution methods influence depressive symptoms in individuals with chronic diseases. Even after transitioning to positive conflict resolution methods, prior negative experiences continued to impact depressive symptoms.

**Data availability statement:** The third-party data underlying the results of this study are publicly available from the KOWEPS website (http://www.koweps.re.kr). To obtain the data, users must create an account, submit a data request, and agree to the data usage terms set by the Korea Institute for Health and Social Affairs (KIHASA). The specific datasets used in this study include survey waves from 2012 to 2022. Interested researchers can fully replicate our study findings by directly obtaining the same datasets from the KOWEPS database and following the methods described in the study. The authors had no special access privileges, and all researchers who meet the KOWEPS data usage requirements can access the data under the same conditions.

**Funding:** The author(s) received no specific funding for this work.

**Competing interests:** The authors have declared that no competing interests exist.

## Introduction

People with chronic conditions have limited activity or persistent pain, which reduces opportunities for activity [1]. These characteristics of chronic disease increase the risk of physical [2], mental [3,4], and psychological distress and lower life satisfaction in people with chronic disease when compared to physically healthy people [5]. Additionally, although stress varies by individual characteristics, patients with higher stress levels due to the general characteristics of chronic disease are more likely to experience mental disorders [6]. Similarly, people with multiple chronic conditions are at a higher risk for depressive symptoms, which is the focus of our study, than people with a single chronic condition [7–9]. Managing depression and its symptom is important because depression can lead to additional risks of social isolation [10], additional psychiatric disease, and suicidal ideation [11]. Therefore, managing the condition of chronic disease patients who are more vulnerable to depressive symptoms is extremely important.

Due to the complexity and unpredictability of the disease, personal management alone has limitations; thus, involving families in disease management and providing psychological support is considered an effective alternative [12]. Family members can improve the condition of chronically ill individuals by offering support and enhancing mutual communication. However, extended periods of sickness can exhaust and frustrate both patients and their families, leading to conflicts and increasing family stress and tension [13,14]. Effective resolution of family conflicts can significantly impact the mental health of both patients and their family members. Each family member may employ various methods of resolving conflicts, including positive methods such as open communication, cooperation, support, and building family resilience [15], or negative methods such as violence, ignoring the problem, or indifference to the family.

The increasing prevalence of depression among patients with chronic diseases in Korea [16] and issues in its management [17] have propelled active research in this direction [18,19]. However, most studies focus on individual behavioral changes [20] or effective interventions by medical staff [21]. Despite the importance of the surrounding environment and family [22], research on how family conflict resolution methods for patients with chronic diseases can affect their depressive symptoms is limited. Therefore, this study aimed to determine whether family conflict resolution among chronically ill patients affects their depressive symptoms and whether changes in conflict resolution, positive or negative, impact depressive symptoms.

## Methods

### Data source and study population

We obtained data from the 7th to 17th waves (2012–2022) of the Korean Welfare Panel Study (KoWePS), an annual longitudinal survey initiated in 2006[23]. This survey aims to assess the living conditions and welfare needs of various demographic groups in relation to changes in work and living environments, age, income levels, and economic activity status. It also seeks to contribute to the development of welfare policies and institutional reforms. The survey is jointly conducted by the Korea Institute for Health and Social Affairs and Seoul National University. The survey employs a proportional systematic stratified cluster sampling method to select a representative sample of households in South Korea. Multiple interviews are conducted within the same household to allow all members aged 15 years or older to complete the questionnaire whenever feasible, thus maximizing participation. The survey is conducted annually with the same participants through face-to-face interviews by the surveyors. However, when the surveyor cannot meet the respondent due to unavoidable circumstances such

as travel or business trips, hospitalization, military service, or prolonged absence from home due to late-night returns or extended trips, limited telephone or proxy response surveys are conducted.

For our study's data, the presence of chronic diseases was reported by participants themselves. Participants were categorized as chronic disease patients if they reported having the condition for less than 3 months, between 3 and less than 6 months, or more than 6 months since the initial diagnosis and initiation of medication. We excluded participants from 2012 to 2022 who were younger than 20 years old, did not have chronic diseases(e.g., seasonal illnesses, which were categorized as non-chronic), were not married or did not have cohabitants, were diagnosed with depression, or showed depressive symptoms in their first entry year in the panel. At baseline, there were a total of 5,816 participants. The average participation period for the panel participants was 5.4 years. The process of participant selection is detailed in Fig 1.

## Measures

The dependent variable was depressive symptoms. The research employed the Center for Epidemiologic Studies Depression Scale (CESD-11) to assess depressive symptoms. The CESD-11, a concise version of the original 20-item scale, is a well-validated self-report screening instrument [24,25]. The CESD-11 has been applied across a wide range of Korean populations, including those with disabilities [26,27]. The total CESD-11 score is computed by summing the scores for all 11 questions and multiplying them by 20/11 [28,29]. A score of 16 or higher indicates the presence of depressive symptoms. Scores falling within the range of 16–20 indicate mild depressive symptoms, scores between 20 and 24 indicate moderate depressive symptoms, and a score of 25 or higher indicates severe depressive symptoms.

In this study, the main independent variable was the methods for managing family conflict, evaluated using the scoring and evaluation method recommended by KoWePS [23]. We evaluated how family members have interacted and resolved conflicts over the past year based on participants' subjective responses to five items. Participants responded to the items using a scale of "Not at all," "Somewhat disagree," "Neutral," "Somewhat agree," and "Strongly agree." Scores are assigned to each item based on the responses, and higher total scores indicate a more positive assessment. For the statements "There are frequent disagreements in my family," "My family members get so angry that they throw things," "My family members criticize each other," and "My family members hit each other," a response of "Not at all" is given 5 points, and a response of "Strongly agree" is given 1 point. For the statement "My family members discuss issues calmly," a response of "Not at all" is given 1 point, and a response of "Strongly agree" is given 5 points. The calculated values are then divided by 5 for evaluation. The reliability of the scale was Cronbach's α = 0.7851. In order to establish categorical thresholds for resolving family conflicts, the responses were grouped into two distinct groups based on the median of the number of responses. Similarly, categorical thresholds were determined for each question based on the median value. Responses exceeding the median were classified as positive approaches to conflict resolution, while those falling below the median were classified as negative approaches.

We controlled for potential confounding variables in our study [30]. Socio-demographic factors included sex, age (20–29/ 30–39/ 40–49/ 50–59/ ≥ 60), region (metropolitan/non-metropolitan areas), household income (based on adjusted gross income below 60% of median as low/60% and above as normal), economic activities (yes/no), and education level (college or higher/high school/middle school or lower). Health-related factors included smoking (yes/no), current alcohol use (yes/no), and self-perceived health status (good/normal/bad). We also included type of chronic disease and family relationship satisfaction as variables.

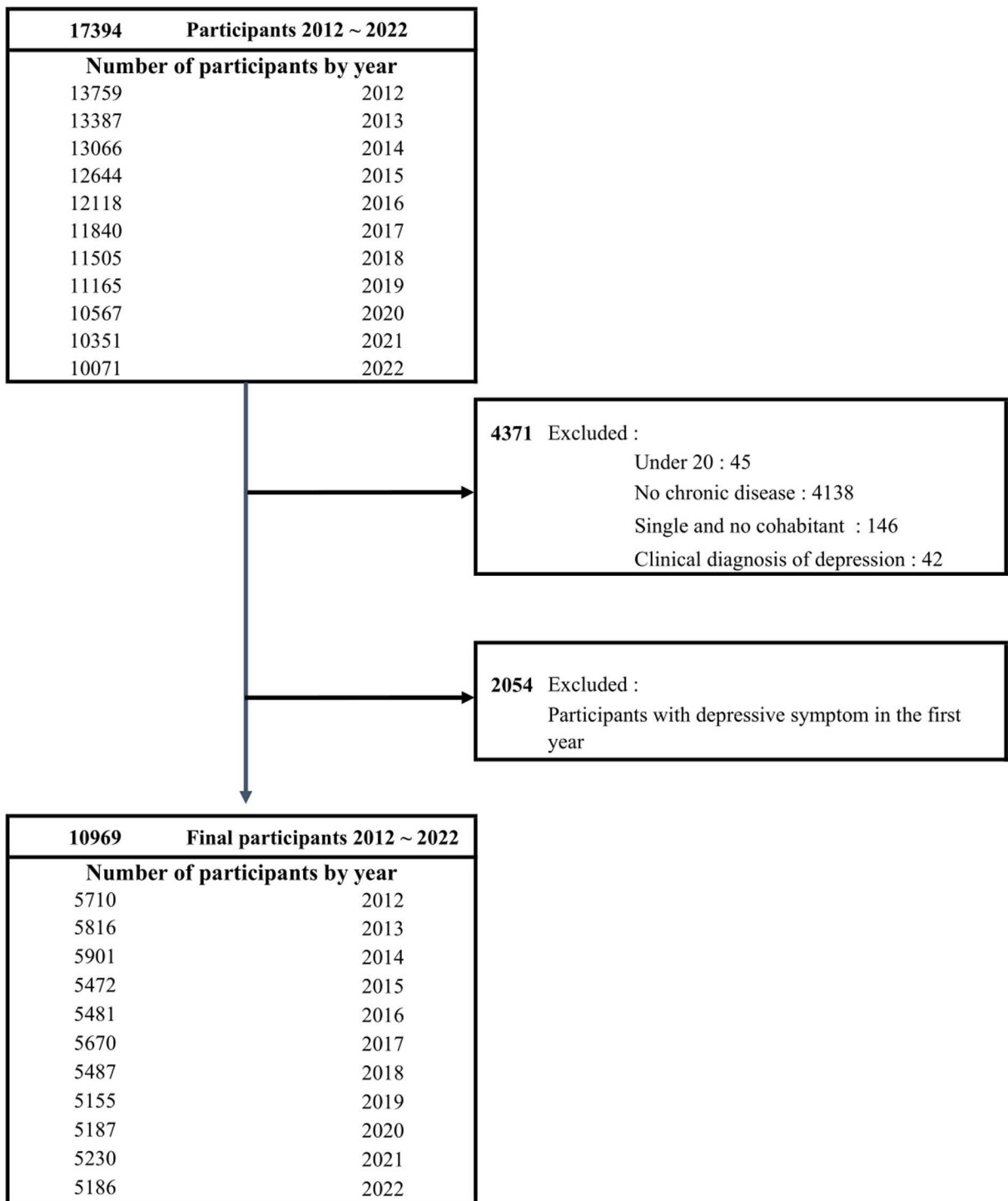

**Fig 1. Flow chart of participant selection.**

Chronic disease data from KoWePS were used to classify the 32 selected chronic diseases into the top 4 most frequently reported primary conditions(excluding depression) and others (hypertension/arthritis/acute conditions/diabetes/other chronic diseases). The variable for

family relationship satisfaction was categorized based on personal satisfaction with family relationships over the past year (dissatisfaction/satisfaction). All responses were self-reported and collected according to the format specified by KoWePS.

## Statistical analysis

Chi-square tests were utilized to assess the overall characteristics of the groups, and generalized estimating equation (GEE) models were applied for the regression analysis of CESD-11 scores, variations in family conflict resolution strategies, and additional covariates. Time variables were grouped into yearly intervals, and unique identifiers were used to recognize recurring participants in the GEE model through the application of an unstructured working correlation matrix. The findings were reported in terms of odds ratios (OR) and confidence intervals (CI). Subgroup analyses were performed to assess the relationships between individual fluctuations in family conflict resolution methods and depressive symptoms, and other pertinent factors. The analyses were performed using SAS software (version 9.4; SAS Institute, Cary, NC, USA), with statistical significance established at a p-value of less than 0.05.

## Results

Table 1 presents baseline characteristics from 2013 for the 5,816 participants included in the analysis. Among them, 32.1% maintained positive family conflict resolution methods, 20.9% transitioned from negative to positive methods, and 18% shifted from positive to negative methods. Additionally, 29% maintained negative methods. Statistically significant differences in depressive symptoms among chronic disease patients were observed based on their family conflict resolution methods.

Table 2 displays the outcomes of the GEE examination concerning the correlation between strategies for resolving family conflicts and depressive symptoms. Model 1 represents the model without adjusting for covariates (negative → negative OR = 1.88, 95% CI = 1.75–2.02). In Model 2, socio-demographic variables were adjusted (negative → negative OR = 1.82, 95% CI: 1.69–1.96). In Model 3, following adjustment for all potential confounding variables, individuals who consistently employed negative conflict resolution methods demonstrated a significantly greater manifestation of depressive symptoms compared to those who consistently utilized positive conflict resolution strategies (negative → negative OR = 1.48, 95% CI = 1.37–1.59).

Table 3 displays GEE results for subgroup analysis, stratified by independent variables. Participants with normal household income showed a higher odds ratio (OR) for depressive symptoms when family conflict resolution methods changed negatively over time (positive → negative OR = 1.31, 95% CI = 1.15–1.48; negative → negative OR = 1.65, 95% CI = 1.46–1.86). Similarly, those engaged in economic activities (positive → negative OR = 1.42, 95% CI = 1.25–1.62; negative → negative OR = 1.62, 95% CI = 1.43–1.84), reporting good health status (positive → negative OR = 1.51, 95% CI = 1.24–1.83; negative → negative OR = 2.07, 95% CI = 1.74–2.46), or expressing dissatisfaction with family relationships (positive → negative OR = 1.25, 95% CI = 1.08–1.45; negative → negative OR = 1.49, 95% CI = 1.30–1.72) were at higher risk of developing depressive symptoms due to negative changes in family conflict resolution.

The findings of the subgroup analysis, stratified based on the variables of interest, are outlined in Fig 2. Family conflict resolution methods were categorized into five distinct items: frequent disagreements, object throwing, collaborative discussions, mutual blame, and incidents of domestic violence. Depressive symptoms increased among individuals who transitioned from negative to positive conflict resolution methods, particularly when collaborative discussions

**Table 1. General characteristics of the baseline population (2012 to 2013).**

| Variables | | Depressive symptoms (CESD-11 ≥ 16) | | | | | | P-value |
|---|---|---|---|---|---|---|---|---|
| | | Total | | No | | Yes | | |
| | | N | % | N | % | N | % | |
| | | **5,816** | **100.0** | **5,112** | **87.9** | **704** | **12.1** | |
| **Family conflict resolution methods** | | | | | | | | <.0001 |
| | Positive → Positive | 1,866 | 32.1 | 1710 | 91.6 | 156 | 8.4 | |
| | Negative → Positive | 1,214 | 20.9 | 1077 | 88.7 | 137 | 11.3 | |
| | Positive → Negative | 1,049 | 18.0 | 913 | 87.0 | 136 | 13.0 | |
| | Negative → Negative | 1,687 | 29.0 | 1412 | 83.7 | 275 | 16.3 | |
| **Sex** | | | | | | | | <.0001 |
| | Male | 2,442 | 42.0 | 2242 | 91.8 | 200 | 8.2 | |
| | Female | 3,374 | 58.0 | 2870 | 85.1 | 504 | 14.9 | |
| **Age** | | | | | | | | <.0001 |
| | 20-29 | 133 | 2.3 | 130 | 97.7 | 3 | 2.3 | |
| | 30-39 | 284 | 4.9 | 275 | 96.8 | 9 | 3.2 | |
| | 40-49 | 557 | 9.6 | 532 | 95.5 | 25 | 4.5 | |
| | 50-59 | 958 | 16.5 | 881 | 92.0 | 77 | 8.0 | |
| | 60≤ | 3,884 | 66.8 | 3294 | 84.8 | 590 | 15.2 | |
| **Region** | | | | | | | | 0.2548 |
| | Metropolitan | 4,348 | 74.8 | 3834 | 88.2 | 514 | 11.8 | |
| | Non-metropolitan | 1,468 | 25.2 | 1278 | 87.1 | 190 | 12.9 | |
| **Household income** | | | | | | | | <.0001 |
| | Normal | 3,370 | 57.9 | 3151 | 93.5 | 219 | 6.5 | |
| | Low | 2,446 | 42.1 | 1961 | 80.2 | 485 | 19.8 | |
| **Economic activity** | | | | | | | | <.0001 |
| | No | 2,498 | 43.0 | 2277 | 91.2 | 221 | 8.8 | |
| | Yes | 3,318 | 57.0 | 2835 | 85.4 | 483 | 14.6 | |
| **Educational level** | | | | | | | | <.0001 |
| | Middle school or lower | 3,606 | 62.0 | 3030 | 84.0 | 576 | 16.0 | |
| | High school | 1,341 | 23.1 | 1252 | 93.4 | 89 | 6.6 | |
| | College or higher | 869 | 14.9 | 830 | 95.5 | 39 | 4.5 | |
| **Smoking** | | | | | | | | 0.0916 |
| | No | 4,949 | 85.1 | 4335 | 87.6 | 614 | 12.4 | |
| | Yes | 867 | 14.9 | 777 | 89.6 | 90 | 10.4 | |
| **Current Alcohol Use** | | | | | | | | <.0001 |
| | No | 3,494 | 60.1 | 2953 | 84.5 | 541 | 15.5 | |
| | Yes | 2,322 | 39.9 | 2159 | 93.0 | 163 | 7.0 | |
| **Health status** | | | | | | | | <.0001 |
| | Bad | 1,895 | 32.6 | 1421 | 75.0 | 474 | 25.0 | |
| | Normal | 1,818 | 31.3 | 1674 | 92.1 | 144 | 7.9 | |
| | Good | 2,103 | 36.2 | 2017 | 95.9 | 86 | 4.1 | |
| **Family relationship** | | | | | | | | <.0001 |
| | Dissatisfaction | 1,157 | 19.9 | 869 | 75.1 | 288 | 24.9 | |
| | Satisfaction | 4,659 | 80.1 | 4243 | 91.1 | 416 | 8.9 | |
| **Chronic disease** | | | | | | | | <.0001 |
| | Hypertension | 1,604 | 27.6 | 1,443 | 90.0 | 161 | 10.0 | |
| | Arthritis | 1,444 | 24.8 | 1,218 | 84.3 | 226 | 15.7 | |
| | Acute diseases | 305 | 5.2 | 281 | 92.1 | 24 | 7.9 | |

*(Continued)*

**Table 1.** (Continued)

| Variables | | Depressive symptoms (CESD-11 ≥ 16) | | | | | | P-value |
|---|---|---|---|---|---|---|---|---|
| | | Total | | No | | Yes | | |
| | | N | % | N | % | N | % | |
| | | **5,816** | **100.0** | **5,112** | **87.9** | **704** | **12.1** | |
| | Diabetes | 632 | 10.9 | 549 | 86.9 | 83 | 13.1 | |
| | Other Chronic diseases | 1,831 | 31.5 | 1621 | 88.5 | 210 | 11.5 | |

**Table 2.** Unadjusted and adjusted modal for depressive symptoms with change of family conflict resolution methods in 2012 to 2022.

| Variables[a] | | Depressive symptoms (CESD-11 ≥ 16) | | | | |
|---|---|---|---|---|---|---|
| | | OR | 95%CI | | | |
| **Family conflict resolution methods** | | | | | | |
| **Model I[a]** | | | | | | |
| | Positive → Positive | 1.00 | | | | |
| | Negative → Positive | 1.23 | (1.15 | – | 1.32) | |
| | Positive → Negative | 1.55 | (1.45 | – | 1.66) | |
| | Negative → Negative | 1.88 | (1.75 | – | 2.02) | |
| **Model II[b]** | | | | | | |
| | Positive → Positive | 1.00 | | | | |
| | Negative → Positive | 1.22 | (1.13 | – | 1.31) | |
| | Positive → Negative | 1.53 | (1.43 | – | 1.64) | |
| | Negative → Negative | 1.82 | (1.69 | – | 1.96) | |
| **Model III[c]** | | | | | | |
| | Positive → Positive | 1.00 | | | | |
| | Negative → Positive | 1.14 | (1.05 | – | 1.22) | |
| | Positive → Negative | 1.34 | (1.24 | – | 1.44) | |
| | Negative → Negative | 1.48 | (1.37 | – | 1.59) | |

[a]Adjusted for Sex, Age;

[b]Additionally adjusted for region, household income, economic activities, education level on the base of Model 1.

[c]Additionally adjusted for smoking, current alcohol use, health status, type of chronic disease, family relationship on the base of Model 2.

(positive → negative OR = 1.2, 95% CI = 1.11–1.30; negative → negative OR = 1.8, 95% CI = 1.65–1.95) and incidents of domestic violence (positive → negative OR = 1.31, 95% CI = 1.20–1.43; negative → negative OR = 1.37, 95% CI = 1.20–1.55) changed from positive to negative or remained negative.

## Discussion

Understanding how to resolve conflicts that may arise during family interactions is important for enhancing the mental health status of people with chronic diseases, as they have a higher risk of developing depression compared to those without chronic diseases. Therefore, this longitudinal study examined the relationship between depression and positive and negative changes in family conflict resolution in patients with chronic diseases in Korea aged over 20.

Our study found an association between depressive symptoms and changes in how patients with chronic diseases resolve family conflicts. A change in the negative approach to resolving conflict and no change in the negative approach were linked to higher odds of experiencing depressive symptoms compared with the positive approach. By contrast, participants who

**Table 3. Subgroup analysis using the generalized estimating equation depressive symptom among chronic patients with family conflict resolution methods in 2012 to 2022.**

| Variables[a] | Depressive symptoms (CESD-11 ≥ 16) | | | | | | | | | | |
|---|---|---|---|---|---|---|---|---|---|---|---|
| | Family conflict resolution methods | | | | | | | | | | |
| | Positive → Positive | Negative → Positive | | | Positive → Negative | | | Negative → Negative | | | |
| | OR | OR | 95% CI | | | OR | 95% CI | | | OR | 95% CI | | |
| **Sex** | | | | | | | | | | | |
| Male | 1.00 | 1.24 | (1.09 | – | 1.42) | 1.44 | (1.25 | – | 1.65) | 1.57 | (1.37 | – | 1.80) |
| Female | 1.00 | 1.09 | (1.00 | – | 1.19) | 1.29 | (1.18 | – | 1.42) | 1.44 | (1.32 | – | 1.58) |
| **Age** | | | | | | | | | | | |
| 20-29 | 1.00 | – | | | | – | | | | – | | | |
| 30-39 | 1.00 | 0.80 | (0.45 | – | 1.44) | 1.11 | (0.58 | – | 2.14) | 1.17 | (0.66 | – | 2.08) |
| 40-49 | 1.00 | 1.36 | (0.89 | – | 2.10) | 1.27 | (0.83 | – | 1.96) | 1.97 | (1.31 | – | 2.95) |
| 50-59 | 1.00 | 1.23 | (0.95 | – | 1.60) | 1.46 | (1.14 | – | 1.88) | 1.96 | (1.55 | – | 2.47) |
| 60 ≤ | 1.00 | 1.12 | (1.04 | – | 1.22) | 1.34 | (1.23 | – | 1.45) | 1.41 | (1.30 | – | 1.53) |
| **Region** | | | | | | | | | | | |
| Metropolitan | 1.00 | 1.17 | (1.07 | – | 1.28) | 1.37 | (1.25 | – | 1.49) | 1.52 | (1.39 | – | 1.66) |
| Non-metropolitan | 1.00 | 1.07 | (0.93 | – | 1.22) | 1.28 | (1.11 | – | 1.48) | 1.38 | (1.19 | – | 1.59) |
| **Household income** | | | | | | | | | | | |
| Normal | 1.00 | 1.13 | (1.00 | – | 1.28) | 1.31 | (1.15 | – | 1.48) | 1.65 | (1.46 | – | 1.86) |
| Low | 1.00 | 1.13 | (1.03 | – | 1.24) | 1.36 | (1.24 | – | 1.49) | 1.38 | (1.25 | – | 1.51) |
| **Economic activity** | | | | | | | | | | | |
| No | 1.00 | 1.17 | (1.07 | – | 1.28) | 1.29 | (1.17 | – | 1.41) | 1.40 | (1.28 | – | 1.53) |
| Yes | 1.00 | 1.06 | (0.93 | – | 1.21) | 1.42 | (1.25 | – | 1.62) | 1.62 | (1.43 | – | 1.84) |
| **Educational level** | | | | | | | | | | | |
| Middle school or lower | – | – | – | – | – | – | – | – | – | – | – | – | – |
| High school | 1.00 | 1.22 | (1.01 | – | 1.46) | 1.21 | (1.00 | – | 1.46) | 1.53 | (1.28 | – | 1.84) |
| College or higher | 1.00 | 1.19 | (0.92 | – | 1.54) | 1.33 | (1.03 | – | 1.72) | 1.84 | (1.43 | – | 2.37) |
| **Smoking** | | | | | | | | | | | |
| No | 1.00 | 1.13 | (1.04 | – | 1.22) | 1.37 | (1.27 | – | 1.48) | 1.48 | (1.36 | – | 1.60) |
| Yes | 1.00 | 1.15 | (0.91 | – | 1.47) | 1.06 | (0.82 | – | 1.36) | 1.41 | (1.11 | – | 1.79) |
| **Current Alcohol Use** | | | | | | | | | | | |
| No | 1.00 | 1.12 | (1.03 | – | 1.21) | 1.31 | (1.21 | – | 1.43) | 1.41 | (1.29 | – | 1.53) |
| Yes | 1.00 | 1.21 | (1.03 | – | 1.42) | 1.47 | (1.25 | – | 1.73) | 1.68 | (1.44 | – | 1.96) |
| **Health status** | | | | | | | | | | | |
| Bad | 1.00 | 1.08 | (0.98 | – | 1.19) | 1.30 | (1.17 | – | 1.43) | 1.28 | (1.16 | – | 1.41) |
| Normal | 1.00 | 1.21 | (1.06 | – | 1.40) | 1.31 | (1.13 | – | 1.51) | 1.49 | (1.31 | – | 1.70) |
| Good | 1.00 | 1.20 | (0.98 | – | 1.46) | 1.51 | (1.24 | – | 1.83) | 2.07 | (1.74 | – | 2.46) |
| **Family relationship** | | | | | | | | | | | |
| Dissatisfaction | 1.00 | 1.10 | (0.93 | – | 1.29) | 1.25 | (1.08 | – | 1.45) | 1.49 | (1.30 | – | 1.72) |
| Satisfaction | 1.00 | 1.14 | (1.05 | – | 1.24) | 1.37 | (1.25 | – | 1.49) | 1.39 | (1.27 | – | 1.52) |
| **Chronic disease** | | | | | | | | | | | |
| Hypertension | 1.00 | – | | | | – | | | | – | | | |
| Arthritis | 1.00 | 1.13 | (0.95 | – | 1.33) | 1.37 | (1.15 | – | 1.64) | 1.46 | (1.24 | – | 1.73) |
| Acute diseases | 100 | 0.86 | (0.53 | – | 1.41) | 1.38 | (0.88 | – | 2.17) | 1.71 | (1.13 | – | 2.57) |
| Diabetes | 1.00 | 1.04 | (0.81 | – | 1.35) | 1.47 | (1.15 | – | 1.89) | 1.45 | (1.15 | – | 1.82) |
| Other Chronic diseases | 1.00 | 1.13 | (1.02 | – | 1.25) | 1.23 | (1.11 | – | 1.36) | 1.45 | (1.31 | – | 1.61) |

[a]Adjusted for all covariates

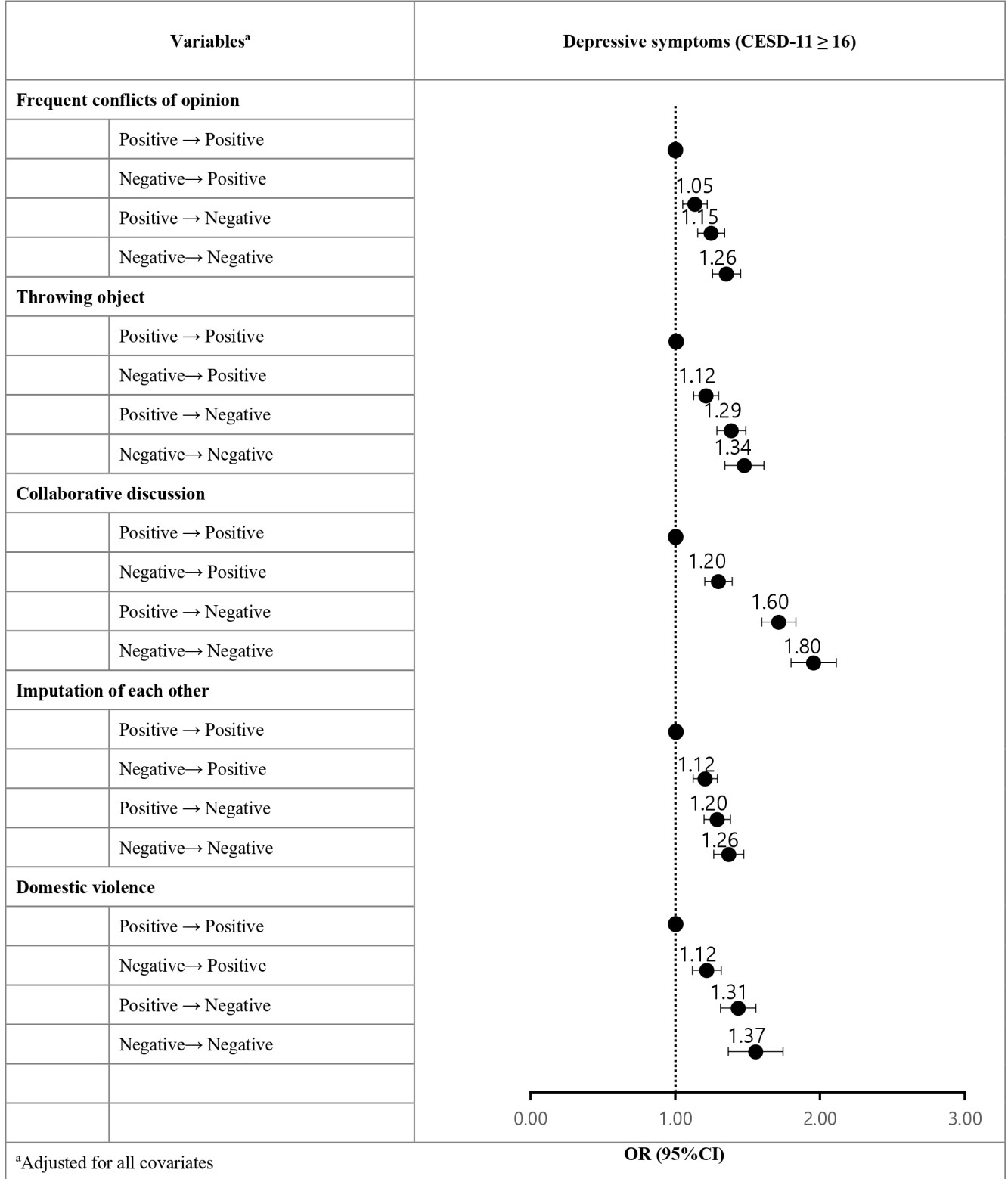

**Fig 2. Sensitivity analysis of depressive symptoms stratified by family conflict resolution methods in 2012 to 2022.**

changed from a negative to a positive method showed lower levels of depressive symptoms than those who continued using a negative method of conflict resolution. However, even among those who adopted positive methods, any prior experience with negative conflict resolution was still associated with depressive symptoms. While transitions from negative to positive methods were linked to lower odds of depressive symptoms compared to remaining in negative conflict styles, the odds ratios remained above one, suggesting that any exposure to negative family conflict experiences contributes to heightened depression risks. Consistent with previous studies, our results indicated that conflict within family relationships influenced depressive symptoms, and addressing conflicts is crucial for managing such symptoms [31, 32].

Our study revealed that economically active participants with chronic diseases faced higher odds of depressive symptoms when their family conflict resolution methods remained persistently negative. This finding aligns with existing research indicating that individuals with chronic diseases are more susceptible to presenteeism [33] and psychological stress in the workplace compared to healthy individuals [34], which can spill over into the home, exacerbating family relationship conflicts [35]. Therefore, addressing mental health issues for individuals with chronic diseases is crucial in both workplace and home settings [36]. This underscores the need for continuous attention and support to improve the overall mental health of individuals with chronic diseases.

Regarding family conflict resolution methods, our study identified frequent differences in opinions, object throwing, collaborative discussions, mutual blame, and incidents of domestic violence. Notably, avoiding collaborative discussions and engaging in object throwing were associated with depressive symptoms. Previous research consistently links family violence [37] or trauma [38] to higher depression risk among individuals, highlighting the need to prevent depression in vulnerable chronic disease patients in violent family environments [39], and to control family conflicts. Additionally, our study findings show that verbal conflict is as strongly linked to an increase in depression and depressive symptoms among patients with chronic diseases as physical conflict [40]. Previous research shows that frequent arguments that include negative language increase depression [41–43], whereas calm and respectful family conversations reduce depression [44, 45]. Across variables, individuals with even a single instance of negative conflict resolution experience exhibited higher levels of depressive symptoms than those who consistently used positive methods. These results reinforce the critical need to prevent negative family conflict altogether and to promote consistent use of positive conflict resolution methods to mitigate depressive symptoms and support mental health in individuals with chronic diseases.

Although the results of this study indicated the need to investigate the relationship between family conflict resolution methods and depressive symptoms in patients with chronic diseases, this study has some limitations. First, this study used the CESD-11 scale to assess depressive symptoms in individuals with chronic diseases. While the CESD-11 is widely used to identify depressive symptoms across a broad range of Korean populations, including individuals with disabilities, its validity specifically for individuals with chronic diseases has not been established. Second, to identify patients with chronic diseases, the type and duration of the disease were checked, and patients without chronic conditions were excluded. However, there may have been cases in which participants were undiagnosed. Third, family conflict resolution methods were assessed through subjective statements; however, similar situations could be interpreted differently by different participants. Fourth, the median was used as a cutoff score to categorize family conflict resolution methods into positive and negative categories; however, this was an unverified criterion. Fifth, identifying causal relationships was not possible as this was a prospective study. Sixth, while the survey primarily adheres to the principle of direct face-to-face interviews, participants who were unable to meet with the surveyors may have completed the survey through telephone or proxy responses. Although this study has some drawbacks, it also has notable strengths. First, it is based on longitudinal data gathered from diverse samples across

Korea. Second, the research highlights the impact of changes in family conflict resolution strategies on depressive symptoms in individuals with chronic disease, showing that negative changes can worsen these symptoms. Additionally, we found that even after transitioning to positive conflict resolution methods, prior negative experiences could still affect depressive symptoms. Last, the study reveals a strong link between resolving verbal conflicts and an increase in depressive symptoms among patients with chronic disease, similar to the effects of physical violence.

## Conclusion

Our study examined the relationship between family conflict resolution methods and depression in individuals over the age of 20 in Korea living with chronic disease. The study found that using negative methods to resolve family conflicts was associated with symptoms of depression. There was a heightened risk of depressive symptoms when negative communication or physical violence was involved. Based on these results, programs and services must be developed to promote healthy conflict resolution for individuals and families, minimize negative change, and facilitate positive change. To enhance the study's scope, future research should incorporate different variables and effective conflict resolution strategies. Furthermore, it is important to compare and validate these findings with those of similar studies conducted in diverse cultural contexts to enhance the applicability of the research outcomes.

## Supporting information

**S1 Table. General characteristics of the baseline population with chronic disease.**
(TIF)

## Acknowledgement

We express our gratitude to the Korea Welfare Panel study for providing longitudinal survey data. Additionally, we thank colleagues from Yonsei University's Health Research Institute for their advice on drafting the manuscript.

## Author contributions

**Conceptualization:** Min Jeong Joo.

**Data curation:** Min Jeong Joo, Jisu Ko, Dan Bi Kim.

**Formal analysis:** Jisu Ko.

**Investigation:** Dan Bi Kim.

**Methodology:** Jae Hyeok Lim.

**Supervision:** Eun-Cheol Park.

**Visualization:** Jae Hyeok Lim.

**Writing – original draft:** Min Jeong Joo.

**Writing – review & editing:** Eun-Cheol Park.

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
