## [Decision Letter · Decision Letter 0]

18 Apr 2024

PONE-D-24-06786The relationship between family conflict resolution methods and depressive symptoms in patients with chronic diseasesPLOS ONE

Dear Dr. Park,

Thank you for submitting your manuscript to PLOS ONE. After careful consideration, we feel that it has merit but does not fully meet PLOS ONE’s publication criteria as it currently stands. Therefore, we invite you to submit a revised version of the manuscript that addresses the points raised during the review process.

We look forward to receiving your revised manuscript.

Kind regards,

Asres Bedaso Tilahune

Academic Editor

PLOS ONE

Journal Requirements:

2. We note that your Data Availability Statement is currently as follows: 

"All relevant data are within the manuscript and its Supporting Information files."

**Additional Editor Comments:**

The study examined an important area of research, but the authors should address the suggested comments before considering publication.

Address all comments raised by both reviewers. In the methods section, use a diagram that is clear and easy for readers to understand how the sample used for the current analyses was selected from the total eligible sample. Explain more detail how the family conflict resolution method was analysed (e.g. how consecutive years were considered to decide whether family conflict resolution is positive/negative?) Clearly label outcome, exposure and confounding variables in the methods section. Also, provide more information on the instruments used to assess some of the variables, such as income (low/normal?), family life, health status, smoking (current/ever?), drinking (alcohol or what?), etc. Make major revisions to the data presented in all tables (N(%), 95%CI, age category?, p-value for age in table 1, include % for all frequencies across all rows, p-value for chronic disease?, be consistent with decimals across all tables (e.g. <.0001, 0.0007), keep journal rule for spacing, table 2 95% CI should be revised). In Table 2, what is the point of reporting the OR of other confounders when your exposure variable (EV) is family conflict resolution method? Instead, present the OR of the EV in different models (Model I (unadjusted), Model II, Model III (fully adjusted)) and report the adjusted variables as a footnote below the table. Delete a figure presented before the discussion. I think the result presented in Table 5 does not add anything to the study as the association between category of family conflict resolution method and depression is significant at almost all levels. Significantly revise the discussion, add more scientific theories to support your argument, use updated literature and include the policy and research implications of the study findings. Revise and update the conclusion accordingly.

Reviewers' comments:

Reviewer's Responses to Questions

**Comments to the Author**

1. Is the manuscript technically sound, and do the data support the conclusions?

Reviewer #1: Yes

Reviewer #2: Yes

2. Has the statistical analysis been performed appropriately and rigorously? 

Reviewer #1: Yes

Reviewer #2: Yes

3. Have the authors made all data underlying the findings in their manuscript fully available?

Reviewer #1: No

Reviewer #2: Yes

4. Is the manuscript presented in an intelligible fashion and written in standard English?

Reviewer #1: Yes

Reviewer #2: No

5. Review Comments to the Author

Reviewer #1: This was a really interesting topic which addressed how family conflict resolve methods may affect depression among chronic disease patients who are likely to be more depressive than those disease-free individuals. They did careful analyses, however I have following questions, hopefully they can be addressed by investigators.

1 there are so many people excluded at baseline, I think a characteristic comparison table is needed to compare those included and excluded

2 why excluded those under 29?

3 is CESD-11 validated in Korean population with chronic diseases ?

4 the scoring method for participants’ responses to five statements are confusing: 1 point for “never “ to 5 point for “always” (line 117), so the higher the total score, the more negative resolve method is, but why the opposite is the definition being stated?

6. What was the plot for on page 13? What is the interpretation of the results?

Reviewer #2: 1. In introduction : how vulnerable people with chronic diseases are to depression, not depression.

2. In Methods : Isn't the participants for people under 30 years old? Line 101 needs to be checked.

3. In Methods : The author should further explain what chronic diseases are.

4. In Methods : Didn't investigate the duration of the prevalence of chronic diseases?

5. In Results : Please organize the tables and Figure in an easy to see. The % is missing in Table 1.

6. In Results : table 5, It would be better to present either the table or the figure, not both.

7. In discussion : I don't understand the sentence. Please organize your thoughts and write them down again. The arguments before and after do not match.

8. In discussion : Please write down the line number.

9. In Conclusion : Please check the sentence and write it down concisely.

10. In Conclusion : Please check the completion of the sentence in the first paragraph.

6. PLOS authors have the option to publish the peer review history of their article (what does this mean? ). If published, this will include your full peer review and any attached files.

**Do you want your identity to be public for this peer review?** For information about this choice, including consent withdrawal, please see our Privacy Policy .

Reviewer #1: No

Reviewer #2: No

---

## [Author Response · Author response to Decision Letter 1]

13 Jul 2024

We were pleased to have the opportunity to revise our paper. In revising our paper, we have carefully considered your comments and suggestions. As instructed, we have attempted to explain the changes made in reaction to all the reviewers’ comments. The reviewers’ comments were very helpful overall, and we appreciate the constructive feedback on our original submission. After addressing the issues raised, we feel the quality of the paper has greatly improved and we hope you agree. Our response to each comment is as follows, and we attach a revision note with the highlighted, revised sections of the manuscript. Again, thank you for the valuable and helpful comments.

---

## [Decision Letter · Decision Letter 1]

27 Dec 2024

PONE-D-24-06786R1The relationship between family conflict resolution methods and depressive symptoms in patients with chronic diseasesPLOS ONE

Dear Dr. Park,

Thank you for submitting your manuscript to PLOS ONE. After careful consideration, we feel that it has merit but does not fully meet PLOS ONE’s publication criteria as it currently stands. Therefore, we invite you to submit a revised version of the manuscript that addresses the points raised during the review process.

**ACADEMIC EDITOR: **

**Thank you for taking the time to thoughtfully revise your paper with the comments of the previous review. Your manuscript has improved greatly. One of the reviewers was "new" to your manuscript, as the previous reviewer was unavailable. That reviewer had some very helpful suggestions that you will see listed below. Please carefully consider these suggestions as you complete your revision. Looking forward to receiving your revised manuscript.**

We look forward to receiving your revised manuscript.

Kind regards,

Ali A. Weinstein, Ph.D.

Academic Editor

PLOS ONE

**Journal Requirements:**

Reviewers' comments:

Reviewer's Responses to Questions

**Comments to the Author**

1. If the authors have adequately addressed your comments raised in a previous round of review and you feel that this manuscript is now acceptable for publication, you may indicate that here to bypass the “Comments to the Author” section, enter your conflict of interest statement in the “Confidential to Editor” section, and submit your "Accept" recommendation.

Reviewer #1: All comments have been addressed

Reviewer #3: (No Response)

2. Is the manuscript technically sound, and do the data support the conclusions?

Reviewer #1: Yes

Reviewer #3: Yes

3. Has the statistical analysis been performed appropriately and rigorously? 

Reviewer #1: Yes

Reviewer #3: Yes

4. Have the authors made all data underlying the findings in their manuscript fully available?

Reviewer #1: Yes

Reviewer #3: No

5. Is the manuscript presented in an intelligible fashion and written in standard English?

Reviewer #1: Yes

Reviewer #3: Yes

6. Review Comments to the Author

**Reviewer #1:**  The authors provided well prepared reply to my questions. They Addressed all my comments well. No further comments now

**Reviewer #3: ** This study examines the impact of family resolution methods on depressive symptoms among adults aged 20 and over in Korea. The authors utilized data from the Korean Welfare Panel Study (KoWePS), analyzing information from 5,816 participants with chronic diseases collected longitudinally from 2012 to 2022 (7th to 17th waves). Notably, participants did not have depressive symptoms upon initial entry into the panel.

I reviewed both the original and revised versions of the manuscript and observed substantial changes that significantly improved the readability and overall quality of the paper. The topic is highly relevant and engaging. Below are some suggestions for further improvement:

1. Provide more information on KoWePS:

o It would be helpful to include additional background on the Korean Welfare Panel Study. What was the study originally designed to assess? Were the same individuals followed over time, or does the dataset involve a rotation of participants from different families or households?

2. Clarify interview methods:

o What specific circumstances prevented face-to-face surveys? Additionally, please indicate the number of participants interviewed via telephone or proxy responses and discuss any potential biases this may introduce.

3. Define chronic conditions:

o The manuscript states: “Participants were reported as chronic if they had the condition for less than three months, between...” This definition is inconsistent with standard definitions of chronic conditions. Chronic diseases typically persist for a longer duration. Please revise or clarify this classification.

4. Justify the age cutoff:

o Why was the age cutoff set at 20 years? Is there a specific rationale for excluding participants aged 18 or 19? Providing an explanation will strengthen the methodology section.

5. Reassess Family Conflict Resolution Measures:

o The assessment of family resolution methods is a significant limitation of the study.

Citations: Add a citation for KoWePS recommendations regarding the evaluation of family conflict resolution.

Content Validity: Many of the statements used (e.g., “My family members criticize each other” or “My family members hit each other”) appear to measure conflict rather than resolution techniques. Please provide justification for using these items as measures of resolution strategies. Have they been validated in previous studies?

6. Explain family conflict groupings:

o The method section lacks details on how the family conflict resolution groups (e.g., “Positive to Positive,” “Negative to Positive,” “Positive to Negative,” “Negative to Negative”) were formed. What time intervals were used (e.g., year-to-year changes or baseline to follow-up)? Ensure this is clearly defined, as it appears prominently in Table 1.

7. Figure formatting and interpretation:

o The figure before the discussion section lacks a title and proper axis descriptions. Include these to improve clarity for readers.

8. Discussion refinement:

o The discussion appropriately highlights that “Negative to Positive” changes in conflict resolution are associated with lower odds of depression compared to “Positive to Negative” or “Negative to Negative.” However, the odds ratios never dropped below one, indicating that any negative family conflict experience increases the odds of depressive symptoms. This underscores the importance of preventing negative family conflict altogether as a key takeaway from the study.

In summary, the manuscript addresses an important and timely topic with significant implications for understanding the relationship between family dynamics and mental health. Incorporating these recommendations will strengthen the study's rigor and impact.

7. PLOS authors have the option to publish the peer review history of their article (what does this mean? ). If published, this will include your full peer review and any attached files.

**Do you want your identity to be public for this peer review?** For information about this choice, including consent withdrawal, please see our Privacy Policy .

Reviewer #1: No

Reviewer #3: **Yes: ** Mamadou Sy

---

## [Author Response · Author response to Decision Letter 2]

11 Jan 2025

Thank you for your great efforts in reviewing our manuscript.

---

## [Editor Report · Decision Letter 2]

15 Jan 2025

The relationship between family conflict resolution methods and depressive symptoms in patients with chronic diseases

PONE-D-24-06786R2

Dear Dr. Park,

We’re pleased to inform you that your manuscript has been judged scientifically suitable for publication and will be formally accepted for publication once it meets all outstanding technical requirements.

Kind regards,

Ali A. Weinstein, Ph.D.

Academic Editor

PLOS ONE
---

## [Editor Report · Acceptance letter]

PONE-D-24-06786R2

PLOS ONE

Dear Dr. Park,

I'm pleased to inform you that your manuscript has been deemed suitable for publication in PLOS ONE. Congratulations! Your manuscript is now being handed over to our production team.

Kind regards,

on behalf of

Dr. Ali A. Weinstein

Academic Editor

PLOS ONE